# Design Wave Height Parameter Estimation Model Reflecting the Influence of Typhoon Time and Space

**Guilin Liu** [1], **Pengfei Xu** [1], **Yi Kou** [2], **Fang Wu** [3], **Yi Yang** [2], **Daniel Zhao** [4] and **Zaijin You** [5,*]

[1] College of Engineering, Ocean University of China, Qingdao 266100, China; liuguilin73@ouc.edu.cn (G.L.); xupengfei@stu.ouc.edu.cn (P.X.)
[2] Dornsife College, University of Southern California, Los Angeles, CA 90007, USA; yikou@usc.edu (Y.K.); yang854@usc.edu (Y.Y.)
[3] Statistics and Applied Probability, University of California Santa Barbara, Santa Barbara, CA 93106, USA; fangwu0703@gmail.com
[4] Department of Mathematics, Harvard University, Cambridge, MA 02138, USA; danielzhao@college.harvard.edu
[5] College of Transportation Engineering, Dalian Maritime University, Dalian 116000, China
* Correspondence: b.you@dlmu.edu.cn; Tel.: +86-15666810220

**Abstract:** Typhoon storm surge disasters are one of the main restrictive factors of sustainable development in coastal areas. They are one of several important tasks in disaster prevention and reduction in coastal areas and require reasonable and accurate calculations of wave height in typhoon-affected sea areas to predict and resist typhoon storm surge disasters. In this paper, the design wave height estimation method based on the stochastic process and the principle of maximum entropy are theoretically advanced, and it can provide a new idea as well as a new method for the estimation of the return level for marine environmental elements under the influence of extreme weather. The model uses a family of random variables to reflect the influence of a typhoon on wave height at different times and then displays the statistical characteristics of wave height in time and space. At the same time, under the constraints of the given observations, the maximum uncertainty of the unobtainable data is maintained. The new model covers the compound extreme value distribution model that has been widely used and overcomes the subjective interference of the artificially selected distribution function—to a certain extent. Taking the typhoon wave height data of Naozhou Observatory as an example, this paper analyzes the probability of typhoon occurrence frequency at different times and the characteristics of typhoon intensity in different time periods. We then calculate the wave height return level and compare it with traditional calculation models. The calculation results show that the new model takes into account the time factor and the interaction between adjacent time periods. Furthermore, it reduces the subjective human interference, so the calculated results of the typhoon's influence on wave height return level are more stable and accurate.

**Keywords:** stochastic process; maximum entropy; typhoon frequency; design wave height; return level

## 1. Research Background

Globally, typhoons (hurricanes) are one of the natural disasters that have caused the greatest losses to human society. With global warming, the frequency of strong typhoons (hurricanes) is increasing, and hydrological events (including rainfall, runoff, evaporation, flood, drought, tide, storm surge, huge wave) caused by typhoons are more serious and frequent than ever (see Figure 1), which has attracted the close attention of scholars in related fields [1–3]. Typhoon Hato brought a Class 10 gale to Hong Kong on 23 August 2017, causing economic losses of USD 1.02 billion and injuring nearly 100 people. Typhoon "Mangkhut" landed in Hong Kong in the early morning of 16 September 2018, injuring more than 200 people and causing huge economic and property losses [4]. In 2019, the global wind king hurricane "Dorian" landed in the Bahamas on 1 September and brought a

catastrophic storm surge to Grand Bahama Island on 2 September, killing at least 50 people and destroying tens of thousands of houses in the Bahamas. The economic loss exceeded seven billion USD [5]. According to the statistics of the Munich Re Group disaster database, in the six years from 2013 to 2018, 36,200 deaths were caused by global hydrological events with an economic loss of 213 billion USD, of which catastrophic hydrologic events accounted for about 56% [6]. In all catastrophic hydrological events, storm surges and huge waves caused by typhoons have severely threatened the survival and development of humankind with their frequent occurrences and formidable destructive power [7–9]. Establishing projects, such as wave and flood prevention seawalls, is an effective way to prevent the destruction of offshore constructions caused by typhoons (hurricanes) from marine dynamic environmental factors (wind, waves, etc.), as well as to reduce the casualties and losses caused by typhoon surge disasters. One of the key technologies for preventing extreme sea conditions while establishing wave and flood prevention seawalls is to reasonably determine the fortification standards. For this reason, it is very important to consider the statistical characteristics of typhoons in time and construct a typhoon-influenced design wave height estimation model that can reflect the characteristics of both time and space factors.

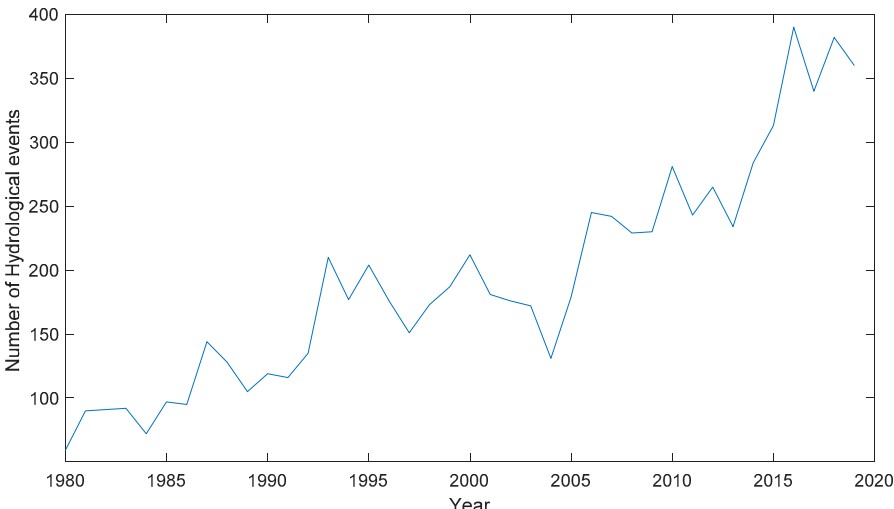

**Figure 1.** Number of Hydrological events causing peril 1980–2019 (https://www.munichre.com/en/risks/natural-disasters-losses-are-trending-upwards/hurricanes-typhoons-cyclones.html, accessed on 27 October 2020).

Design wave height is an important part of the calculation of seawall height in the coastal protection, mitigation, and prevention of typhoon surge disasters [10–12]. For marine engineering to prevent typhoon storm surge disasters, it is of great significance to reasonably and accurately determine the multi-year distribution law of the wave height in the typhoon-affected area and obtain the design value of the corresponding wave return level. In the early design wave height estimation models, marine environmental factors, such as wave height and water level, were simply regarded as random variables for extreme probability analysis. Young et al. analyzed >20 years of global altimeter data, subdivided into four years, and used a Gumbel distribution to analyze the extreme value of each time period [13]. Wang Liping used Gumbel, Weibull, and Pearson-III distributions to analyze the measured wave height data of Binhai Nuclear Power Station from 1963 to 1989 and obtained the multi-year wave height values of the area under three methods [14]. Under several regional wave climatic conditions of the east and west coast of the United States, Neary et al. used the two common extreme value analysis models from the Gumbel distribution of the annual extreme value method (AM) and the generalized Pareto distribution (GPD) of the peak over threshold (POT) method to do the analysis on the extreme sea conditions characterized by wave conditions from the third-generation spectral wave

model. By using quantile-quantile plots to determine the threshold for GPD distribution fitting is indeed a great method to avoid user deviation [15,16]. Clayton et al. proposed a change method for regional frequency analysis. Based on the POT method, the threshold value was selected as the 98th percentile of the wave height, and GPD was used for data fitting; through this, he analyzed the extreme waves on the Pacific coast of Canada [17]. S. CAIRES and A. STERL, based on the European Center for Medium-Range Weather Forecasts (ECMWF) 40-yr Re-Analysis (ERA-40) data, used the POT method to estimate the global wind speed and a hundred-year return level of effective wave height. The 93 and 97 percentiles of the data were selected as the thresholds, and the exponential and GPD distributions were compared. The results showed that the exponential distribution could better fit the existing data [18]. Mazas et al. studied the method of determining extreme wave height through GPD modeling and showed that a higher threshold would improve the rationality of model extrapolation [19]. It is appropriate to select a threshold as large as possible and directly model the extreme tail values, but GPD cannot fit the peak and right tail of over the threshold distribution at the same time, which limits GPD's analysis ability on the overall sample distribution. Simultaneously, the determination of the threshold value is also a difficult task, and artificially selecting a threshold on the fixed percentile of the data is a relatively crude method, which is easily affected by human bias. With the deepening of the research, Liu Defu and Ma Fengshi put forward the compound extreme value distribution theory, which is suitable for the typhoon-affected sea area, extends the one-dimensional compound extreme value distribution to multi-dimensional space [20–24], and has been widely used in engineering. Dong Sheng et al. proposed a Poisson two-dimensional Gumbel logistic distribution model and applied it to the probability and statistical analysis of typhoon surge [25]. The above methods for estimating the design wave height assume that the typhoon wave height data conforms to a certain distribution pattern and use the existing observation data to estimate the parameters in the distribution function. Most of the selected distribution functions can pass the test, but the results of different models are quite distinctive from each other. In 1957, Jaynes proposed the principle of maximum entropy [26]. The essence of maximum entropy is to constrain the known information while maintaining the maximum uncertainty of the unknown information. To a certain extent, it can avoid the influence of a priori and human factors when analyzing the probabilistic characteristics of marine environmental elements such as wave heights. In the 1990s, Xu Delun and others proposed the Maximum-entropy distribution to reduce the a priori factors in estimating marine environmental design parameters [27]. Wang Liping and others combined the compound extreme value theory and the principle of maximum entropy and proposed a new model of design wave height estimation [28]. Liu Guilin et al. derived the joint maximum entropy distribution function of wave height and wave period with the help of the structure of the Copula function to calculate the wave height return level [29]. Following in-depth research, scholars have proposed a multi-dimensional composite maximum entropy model for the joint design of multiple marine environmental elements [30–32] so that the calculation results are more in line with the actual engineering and natural state.

The above design wave height estimation model, whether it is an extreme value distribution, compound extreme value distribution, one-dimensional, or multi-dimensional, they all use typhoon frequency and wave height as random variables to solve the return period level. In fact, marine environmental factors such as typhoon frequency and wave height are essentially stochastic processes, which are constantly changing through time. Treating them as stochastic processes, we establish models and analyze dynamic characteristics changing through time. Thus, we can more accurately describe the state of the project and the natural state of the marine environment elements, such as typhoon wave height, to make the return level typhoon wave height calculation more in line with engineering requirements. To analyze typhoon wave height distributions, carry out more accurate and effective storm surge disaster prevention, and improve defense projects in coastal engineering, we use stochastic process theory to study the internal mechanisms and statistical distributions

of typhoon wave heights, analyze typhoon intensity and typhoon-affected wave height distributions at different times, reflect internal statistical characteristics at a deeper level, and select appropriate distribution functions; all the while avoiding subjective interference of artificially selected distribution functions.

In this paper, based on the Maximum Entropy Principle, no other restrictive assumptions about wave height are made, thus avoiding the a priori factors of the assumed distribution function. At the same time, based on the stochastic process theory and the principle of maximum entropy, this paper uses a cluster of random variables instead of a limited number of random variables to study the statistical characteristics of typhoon wave heights and a new model of typhoon influence wave height distribution is constructed, which reflects the characteristics of time and space. We analyze the probability of typhoon frequency in different time periods and the wave height distribution law under its influence and show that the features of design wave height in temporal and spatial dimensions (while covering the compound extreme value distribution model) that has been widely used and overcomes the subjective interference of artificially selected distribution functions. Based on the Naozhou typhoon and wave height data, this paper provides a model for calculating the return level of typhoon wave heights considering time, with an in-depth analysis of the probability of typhoon occurrence in different seasons and months. It also provides the distribution of design wave heights under the influence of typhoons and calculates the needed height of seawalls in coastal protection projects in typhoon-prone areas. Through these methods, we present more effective and accurate design parameter guidance for preventing and resisting typhoon surge disasters.

## 2. Model Construction

In order to consider the changes in typhoon wave height in time and space, this paper adds the time factor on the basis of the traditional model, using a random process to construct a time-varying typhoon frequency probability model and design wave height distribution pattern under the influence of typhoon to describe frequency and wave height in more detail.

**Theorem 1.** *Set $X(t, \varsigma)$ to wave height at time t, for wave height $\varsigma$ at time t, abbreviated as $X(t)$. $g(x, t)$, $G(x, t)$ represents, respectively, its distribution function and probability density function. Suppose $N(t, k)$ is expressed as the occurrence of typhoon k within the time period of $(0, t)$, the value is non-negative and independent of the wave height, which is simply written as $N(t)$, its probability distribution law at time t is:*

$$\begin{pmatrix} 0 & 1 & \cdots & k \\ p_0(t) & p_1(t) & \cdots & p_k(t) \end{pmatrix} \tag{1}$$

The general formula of the typhoon wave height distribution function based on the stochastic process is:

$$F(x, t) = \sum_{k=0}^{\infty} p_k(t) \cdot k \cdot \int_1^{x(t)} G^{k-1}(u, t) g(u, t) du \tag{2}$$

Particularly, when the frequency of typhoon occurrence $N(t)$ assumes the Poisson process with $\lambda t$, and wave height $X(t)$, taking the maximum entropy distribution, (2) becomes:

$$F(x, t) = \sum_{k=0}^{\infty} \frac{(\lambda t)^k}{k!} e^{-\lambda t} \int_0^{x(t)} \left[ \alpha(u, t)^\gamma \exp(-\beta(u, t)^\xi) \right]^k du \tag{3}$$

where $\lambda$ is the number of typhoons, and $\alpha$, $\beta$, $\gamma$, $\xi$ are the parameters given by the constraints and boundary conditions.

**Proof.** According to engineering practicality, the frequency of typhoon occurrence $N(t)$ has the following statistical characteristics:

(1) The probability of occurrence of more than one typhoon in a sufficiently small time interval is very small, and the frequency of typhoon occurrence $N(t)$ takes a non-negative integer value, with $N(0) = 0$.

(2) The number of occurrences of typhoons in any two non-overlapping time intervals is independent of each other, if $t_1 < t_2$, then $N(t_1) < N(t_2)$, $N(t_1 t_2) = N(t_2) - N(t_1)$ is the number of occurrences in the time period $(t_1, t_2)$.

(3) The number of typhoon occurrence times within the time $(t_1, t_2]$ is only related to the length of the time interval $(t_2 - t_1)$, and is independent to the time $t_1$. $N(t)$ is the continuous step function on the right for $[0, \infty)$. $\square$

It can be seen from the above statistical properties that the number of typhoon occurrences in a certain period of time $N(t)$ is a counting process, assuming it takes the Poisson process with $\lambda t$, $\lambda t = E[X(t)]$ represents the average number of typhoon occurrences within the time $t$.

Recorded $F(x, t)$ as the distribution function of the wave height at the moment of $t$ under the influence of the typhoon, then:

$$
\begin{aligned}
F(x,t) &= P(X(t) \leq x) = P\left( \bigcup_{k=0}^{\infty} \{X(t) \leq x \cap N(t) \leq k\} \right) \\
&= \sum_{k=0}^{\infty} P(X(t) \leq x | N(t) = k) \cdot P(N(t) = k) \\
&= \sum_{k=0}^{\infty} p_k(t) P(X(t) \leq x | N(t) = k) \\
&= p_0(t) \cdot G(x,t) + \sum_{k=1}^{\infty} p_k(t) P(X(t) \leq x | N(t) = k)
\end{aligned}
\tag{4}
$$

in which:

$$
\begin{aligned}
P(X(t) \leq x | N(t) = k) &= P\left( \bigcup_{i=1}^{k} \{X(t) \leq x\} \cap \left\{ \underset{1 \leq j \leq k}{Max}\, \xi_j(t) = \xi_i(t) \right\} \Big| N(t) = k \right) \\
&= \sum_{i=1}^{k} P\left( \{X(t) \leq x\} \cap \left\{ \underset{1 \leq j \leq k}{Max}\, \xi_j(t) = \xi_i(t) \right\} \Big| N(t) = k \right)
\end{aligned}
\tag{5}
$$

Assuming $\xi_i(t) = \xi_1(t)$, and set the first data in the data column to be the largest, then:

$$
\begin{aligned}
P(X(t) \leq x | N(t) = k) &= k P\left( \{X(t) \leq x\} \cap \left\{ \underset{1 \leq j \leq k}{Max}\, \xi_j(t) = \xi_1(t) \right\} \Big| N(t) = k \right) \\
&= k \cdot P(\xi_1(t) < x, \xi_1(t) > \xi_j(t), j = 2, 3, \cdots k | N(t) = k) \\
&= k \cdot E\left\{ \prod_{i=1}^{n} I_{\{\xi_i(t) < x_i\}}(\omega) I_{\{\xi_1(t) > \xi_j(t), j=2,3,\cdots k\}}(\omega) | N(t) = k \right\} \\
&= k \cdot E\left\{ \prod_{i=1}^{n} I_{\{\xi_i(t) < x_i\}}(\omega) \Big| \xi_1(t) = k \right\} \\
&= k \cdot E\left\{ \prod_{i=1}^{n} I_{\{\xi_i(t) < x_i\}}(\omega) G^{i-1}(u, t) \right\} \\
&= k \cdot \int_{-\infty}^{x(t)} G^{k-1}(u, t) g(u, t) du
\end{aligned}
\tag{6}
$$

in which: $I_A = \begin{cases} 1, x \in A \\ 0, x \notin A \end{cases}$ is the indicative function on $A$.

$$
\begin{aligned}
F(x,t) &= p_0(t) \cdot G(x,t) + \sum_{k=1}^{\infty} p_k(t) \cdot k \cdot \int_1^{x(t)} G^{k-1}(u,t) g(u,t) du \\
&= \sum_{k=0}^{\infty} p_k(t) \cdot k \cdot \int_0^{x(t)} G^{k-1}(u,t) g(u,t) du
\end{aligned}
\tag{7}
$$

When the frequency of the typhoon $N(t)$ takes the Poisson process with parameter $\lambda t$, the distribution function is

$$p_k(t) = \frac{(\lambda t)^k}{k!} e^{-\lambda t}, k = 0, 1, 2 \ldots \tag{8}$$

Suppose the wave height $X(t)$ conforms to the maximum entropy distribution [27], and its probability density function is:

$$g(x, t) = \alpha(x, t)^\gamma e^{-\beta(x, t)^\xi} \tag{9}$$

The corresponding distribution function is:

$$G(x, t) = \int_0^{x_t} \alpha(u, t)^\gamma e^{-\beta(u, t)^\xi} du \tag{10}$$

Wherein $\alpha, \beta, \gamma, \xi$ are the parameters given by the constraints and boundary conditions. Under given constraint conditions Euler equations are used [33]. Substituting Equations (8) and (10) into Equation (2), we get:

$$
\begin{aligned}
F(x, t) &= \sum_{k=0}^\infty P_k(t) \cdot k \cdot \int_{-\infty}^{x_t} G^{k-1}(u, t) g(u, t) du \\
&= \sum_{k=0}^\infty \frac{(\lambda t)^k}{k!} e^{-\lambda t} \cdot k \cdot \int_0^{x_t} G^{k-1}(u, t) g(u, t) du \\
&= \sum_{k=0}^\infty \frac{(\lambda t)^k}{k!} e^{-\lambda t} \cdot \int_0^{x_t} k \cdot G^{k-1}(u, t) d(G(u, t)) \\
&= \sum_{k=0}^\infty \frac{(\lambda t)^k}{k!} e^{-\lambda t} \cdot \int_0^{x_t} d\left(G^k(u, t)\right) \\
&= \sum_{k=0}^\infty \frac{(\lambda t)^k}{k!} e^{-\lambda t} \cdot G^k(x_t) \\
&= \sum_{k=0}^\infty \frac{(\lambda t)^k}{k!} e^{-\lambda t} \int_0^{x_t} \left[\alpha(u, t)^\gamma \exp(-\beta(u, t)^\xi)\right]^k du
\end{aligned}
\tag{11}
$$

End of theorem proof.
In practical applications, let:
$$F(x, t) = R \tag{12}$$

where $R = 1 - P$, and $P$ are the design frequency. Return period may be expressed as:

$$T = \frac{1}{P} = \frac{1}{1 - R} \tag{13}$$

Through Formulas (11)–(13), the design value of wave height under the influence of typhoons at any time in different return periods can be calculated.

## 3. Ocean Engineering Calculation Case

### 3.1. Probability Analysis of Typhoon Frequency

Naozhou Observation Station is located on Naozhou Island, Zhanjiang City (21°16′ N, 110°22′ E). Due to the special geographical location of Naozhou Station within the Leizhou Peninsula (21°15′ N~21°20′ N, 109°22′ E~110°27′ E). Among the tropical cyclones that have a great impact on the island in history, few have landed in the west of the Naozhou. Most tropical cyclones make landfall to the south and east of Naozhou. The number of typhoons tends to increase from May to September. According to the typhoon data of the tropical cyclone data center of the China Meteorological Administration (http://tcdata.typhoon.org.cn/dlrdqx_zl.html, accessed on 18 December 2020), the frequency of typhoons and the corresponding Poisson process intensity in Naozhou from June to September in 1990–2016 are listed in Table 1.

**Table 1.** Frequency of typhoons and Poisson intensity $\lambda$.

| Years | Frequency | $\lambda$ | Years | Frequency | $\lambda$ | Years | Frequency | $\lambda$ |
|-------|-----------|-----------|-------|-----------|-----------|-------|-----------|-----------|
| 1990 | 3 | 0.75 | 1999 | 5 | 1.25 | 2010 | 1 | 0.25 |
| 1991 | 4 | 1 | 2000 | 2 | 0.5 | 2011 | 2 | 0.5 |
| 1992 | 3 | 0.75 | 2001 | 3 | 0.75 | 2012 | 3 | 0.75 |
| 1993 | 5 | 1.25 | 2002 | 3 | 0.75 | 2013 | 4 | 1 |
| 1994 | 3 | 0.75 | 2003 | 4 | 1 | 2014 | 2 | 0.5 |
| 1995 | 3 | 0.75 | 2005 | 2 | 0.5 | 2015 | 2 | 0.5 |
| 1996 | 4 | 1 | 2006 | 2 | 0.5 | 2016 | 2 | 0.25 |
| 1997 | 2 | 0.5 | 2008 | 3 | 0.75 | | | |
| 1998 | 1 | 0.25 | 2009 | 4 | 1 | | | |

According to the measured data, only three typhoons occurred in May from 1990 to 2016, and none occurred in May in other years. Therefore, assuming there will be no typhoons in May, consider the probability of a total of one typhoon from June to September. In the case of the 2012 typhoon as an example, from Table 1, we obtain the Poisson intensity $\lambda = 0.75$, the time interval is one month, $t_1$ represents May, $t_5$ as September, and since the typhoon has an independent increment, by Formula (4), there are:

$$
\begin{aligned}
&P(N(t_1) = 0, N(t_5) = 1) \\
&= P(N(t_1) = 0, N(t_5) - N(t_1) = 1) \\
&= P(N(t_1) = 0)P(N(t_2 t_5) = 1) \\
&= \frac{(0.75 \times 1)^0}{0!} e^{-0.75 \times 1} \cdot \frac{(0.75 \times 4)^1}{1!} e^{-0.75 \times 4} \approx 0.0706
\end{aligned}
\tag{14}
$$

Therefore, it can be concluded that when there is no typhoon in May, and the probability of one typhoon occurring from June to September is 7.06%. Thus, the probability of no typhoons in May from 1990 to 2016, while the number of occurrence for typhoons of 1, 2, 3, 4, and 5 times from June to September are shown in Table 2.

From the data in Table 2, it can be seen that the possibility of 2–3 typhoon occurrences in the Naozhou area is high, and the probability is mostly >10%, which is basically in line with the average number of typhoons that occurred in Naozhou from June to September during 1990 to 2016. Taking the calculation results in 2012 as an example, near the middle of the frequency, the occurrence probability of a typhoon is the highest and decreases symmetrically to both sides. Therefore, it is unlikely that it will become larger or smaller. The occurrence of 2–3 typhoons from June to September of that year is the most likely result. The more (or less) typhoon occurrences there are in May, the less likely they are, by this estimation.

According to the calculation results in Table 2, the most likely occurrence frequency of typhoons from June to September each year can be obtained, as shown in Table 3. The data in Tables 1 and 3 are used to draw a line chart of the frequency of typhoons in the corresponding year, as shown in Figure 2.

**Table 2.** Probability of each occurrence frequency of typhoon from June to September.

| Years | Probability of Typhoon Frequency (%) | | | | |
|---|---|---|---|---|---|
| | 1 Time | 2 Times | 3 Times | 4 Times | 5 Times |
| 1990 | 7.06 | 10.58 | 10.58 | 7.94 | 4.76 |
| 1991 | 2.7 | 5.39 | 7.19 | 7.19 | 5.75 |
| 1992 | 7.06 | 10.58 | 10.58 | 7.94 | 4.76 |
| 1993 | 0.97 | 2.41 | 4.02 | 5.03 | 5.03 |
| 1994 | 7.06 | 10.58 | 10.58 | 7.94 | 4.76 |
| 1995 | 7.06 | 10.58 | 10.58 | 7.94 | 4.76 |
| 1996 | 2.7 | 5.39 | 7.19 | 7.19 | 5.75 |
| 1997 | 16.42 | 16.42 | 10.94 | 5.47 | 2.19 |
| 1998 | 28.65 | 14.33 | 4.78 | 1.19 | 0.24 |
| 1999 | 0.97 | 2.41 | 4.02 | 5.03 | 5.03 |
| 2000 | 16.42 | 16.42 | 10.94 | 5.47 | 2.19 |
| 2001 | 7.06 | 10.58 | 10.58 | 7.94 | 4.76 |
| 2002 | 7.06 | 10.58 | 10.58 | 7.94 | 4.76 |
| 2003 | 2.7 | 5.39 | 7.19 | 7.19 | 5.75 |
| 2005 | 16.42 | 16.42 | 10.94 | 5.47 | 2.19 |
| 2006 | 16.42 | 16.42 | 10.94 | 5.47 | 2.19 |
| 2008 | 7.06 | 10.58 | 10.58 | 7.94 | 4.76 |
| 2009 | 2.7 | 5.39 | 7.19 | 7.19 | 5.75 |
| 2010 | 28.65 | 14.33 | 4.78 | 1.19 | 0.24 |
| 2011 | 16.42 | 16.42 | 10.94 | 5.47 | 2.19 |
| 2012 | 7.06 | 10.58 | 10.58 | 7.94 | 4.76 |
| 2013 | 2.7 | 5.39 | 7.19 | 7.19 | 5.75 |
| 2014 | 16.42 | 16.42 | 10.94 | 5.47 | 2.19 |
| 2015 | 16.42 | 16.42 | 10.94 | 5.47 | 2.19 |
| 2016 | 16.42 | 16.42 | 10.94 | 5.47 | 2.19 |

**Table 3.** The most likely frequency of typhoons.

| Years | Frequency of Typhoons | Years | Frequency of Typhoons | Years | Frequency of Typhoons |
|---|---|---|---|---|---|
| 1990 | 2–3 | 1999 | 4–5 | 2010 | 1 |
| 1991 | 4 | 2000 | 1–2 | 2011 | 1–2 |
| 1992 | 2–3 | 2001 | 2–3 | 2012 | 3 |
| 1993 | 5 | 2002 | 2–3 | 2013 | 3–4 |
| 1994 | 2–3 | 2003 | 3–4 | 2014 | 1–2 |
| 1995 | 2–3 | 2005 | 1–2 | 2015 | 1–2 |
| 1996 | 3–4 | 2006 | 1–2 | 2016 | 1–2 |
| 1997 | 1–2 | 2008 | 2–3 | | |
| 1998 | 1 | 2009 | 3–4 | | |

Comparing Tables 1 and 3, and considering the time distribution of typhoon occurrence frequency (the influence of typhoon occurrence in May), the obtained probability analysis result of typhoon occurrence agrees with the actual occurrence frequency of typhoons. In the setting of a larger frequency, the probability analysis results of the typhoon occurrence are completely in line with the actual frequency of the typhoon. Therefore, we can reasonably predict the probability of typhoon frequency through the time distribution of typhoon occurrence, especially with the intensity characteristics of the typhoon frequency period, which provides a basis for the forecast and warning of typhoon activities. At the same time, it can provide more accurate parameter guidance for the calculation of the return level of the wave height in the sea area affected by the typhoon.

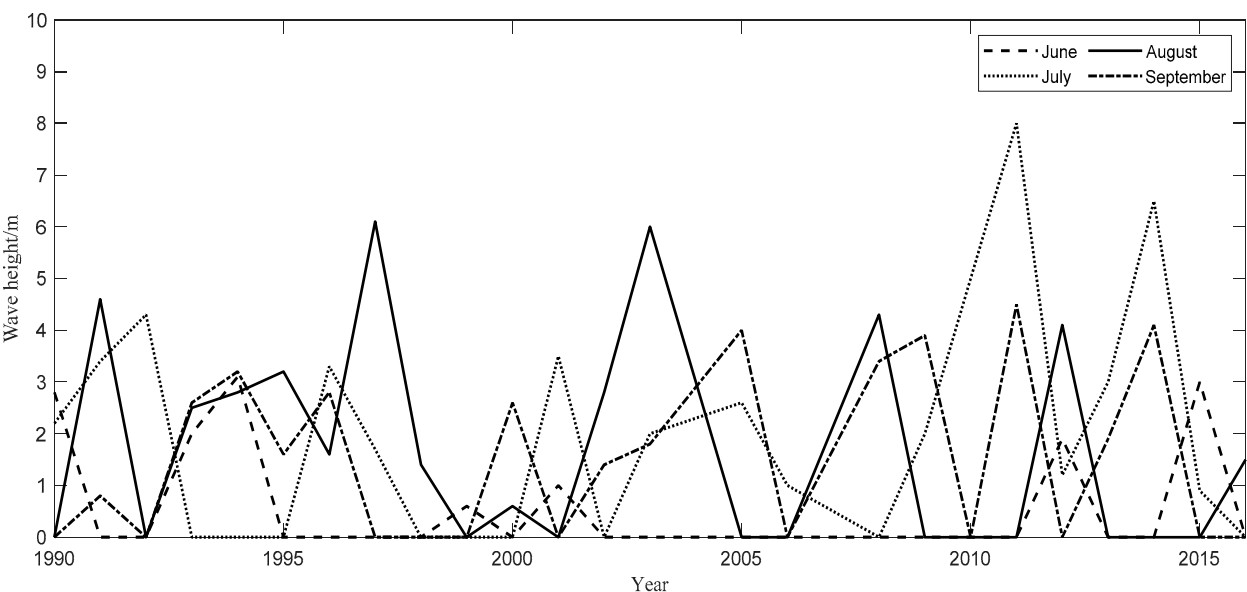

**Figure 2.** The wave height data scatter plot.

### 3.2. Calculation of Design Wave Height in the Sea Area Affected by Typhoon

This paper selects the measured wave height data of the Zhanjiang Naozhou marine environment monitoring station from 1990 to 2016 (missing 2004 and 2007) for analysis. Figure 2 shows the 25-year significant wave height data set from June to September of 1990 to 2016 (missing 2004 and 2007). A value of zero indicates no typhoon in the given month of that year.

This paper uses annual extreme value sampling methods for extreme value statistical analysis of 25-year wave height data. The Gumbel, Weibull, Pearson III, and Maximum Entropy distributions were selected as statistical analysis models. To perform diagnostic tests on the selected samples, Figures 3–6 show four sets of diagnostic test plots, empirical distribution map, quantile map, return level map, and density histogram. In the empirical distribution and quantile diagrams, most of the data points are distributed on and around the line, but the Weibull distribution fits poorly. The return level diagram shows that the data points in the figure are basically distributed on the return level curve of the model. The fitting result of the Gumbel distribution is the best; the density curve intuitively illustrates the distribution of the data. However, the histogram shows that the observation data points have a better fitting result with the Pearson-III distribution and the maximum entropy distribution model. The diagnostic test diagram of each model shows that these four models can be used as the estimation model of the wave height data, but which one to choose needs to be further tested.

Table 4 lists the test results and parameters from the K-S method towards the annual extreme wave height data of the Gumbel, Weibull, Pearson-III, and maximum entropy distributions. Since $D_n < D_0$ (0.05), the null hypothesis of each distribution is passed. The test value of the maximum entropy distribution model is the smallest, which indicates that it has the best fitting degree to the wave height data. The calculation results of design wave height for an n-year (na means the return period is n years or once in n years, and the corresponding frequency is (100/n)%) return period of each distribution model are shown in Table 5.

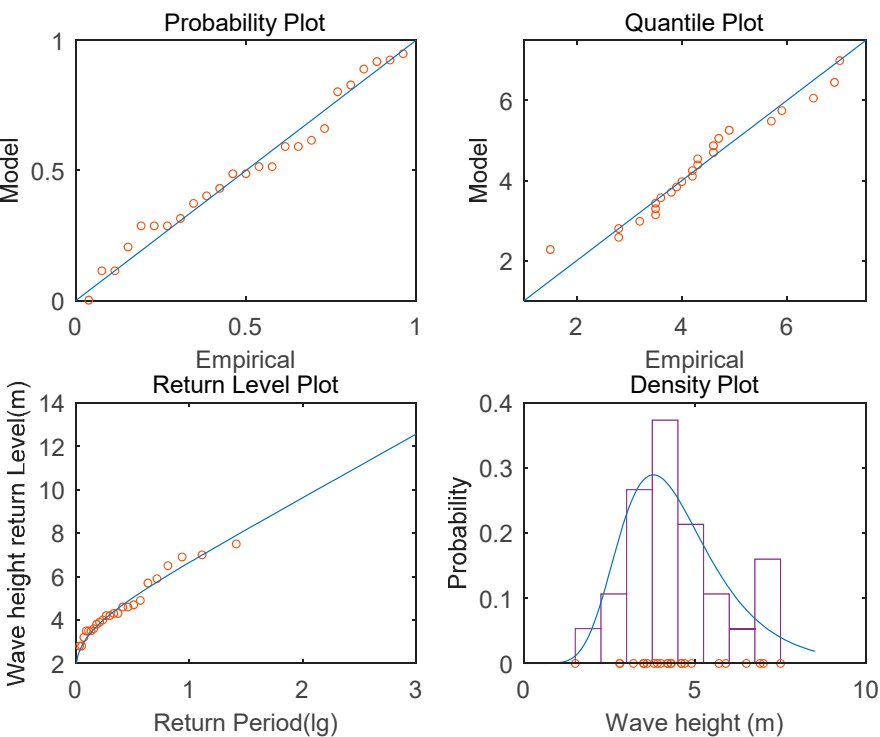

**Figure 3.** Inspection diagram of annual extreme wave height data of Gumbel distribution.

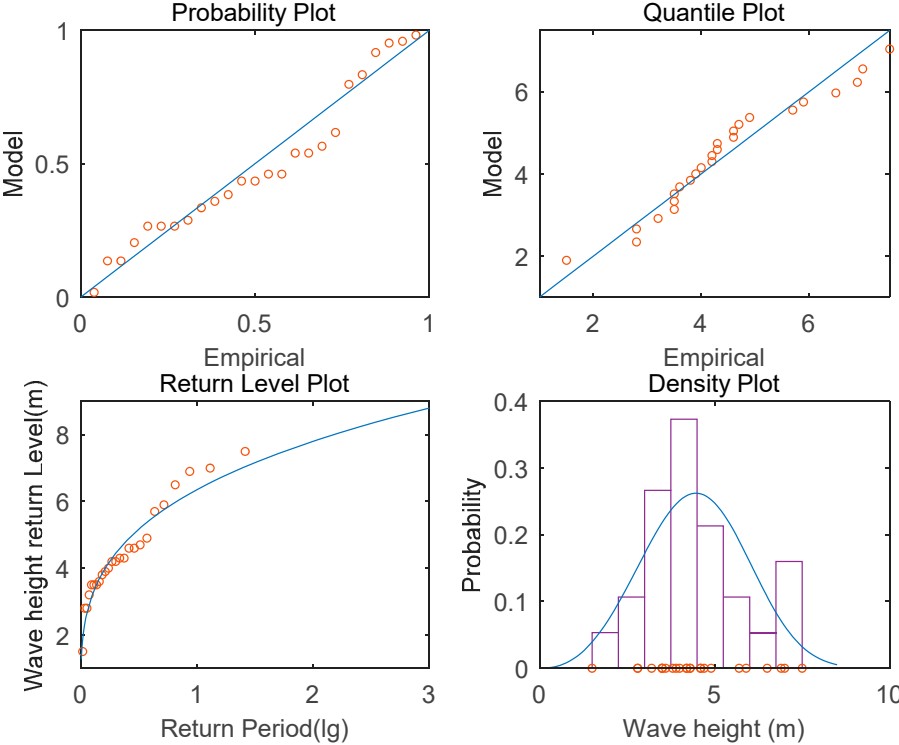

**Figure 4.** Inspection diagram of annual extreme wave height data of Weibull distribution.

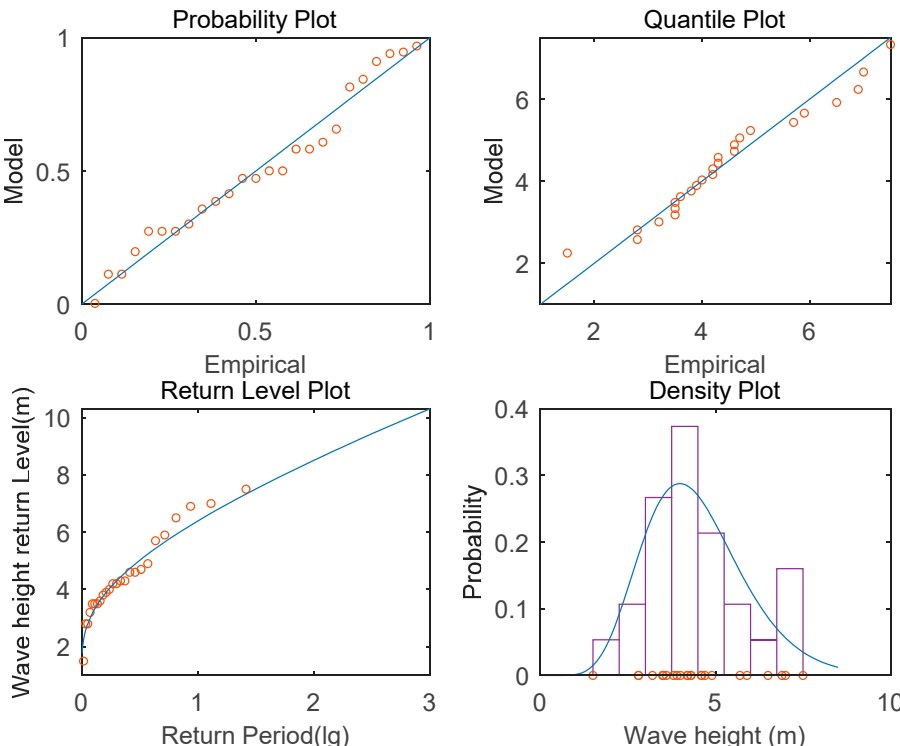

**Figure 5.** Inspection diagram of annual extreme wave height data of Pearson-III distribution.

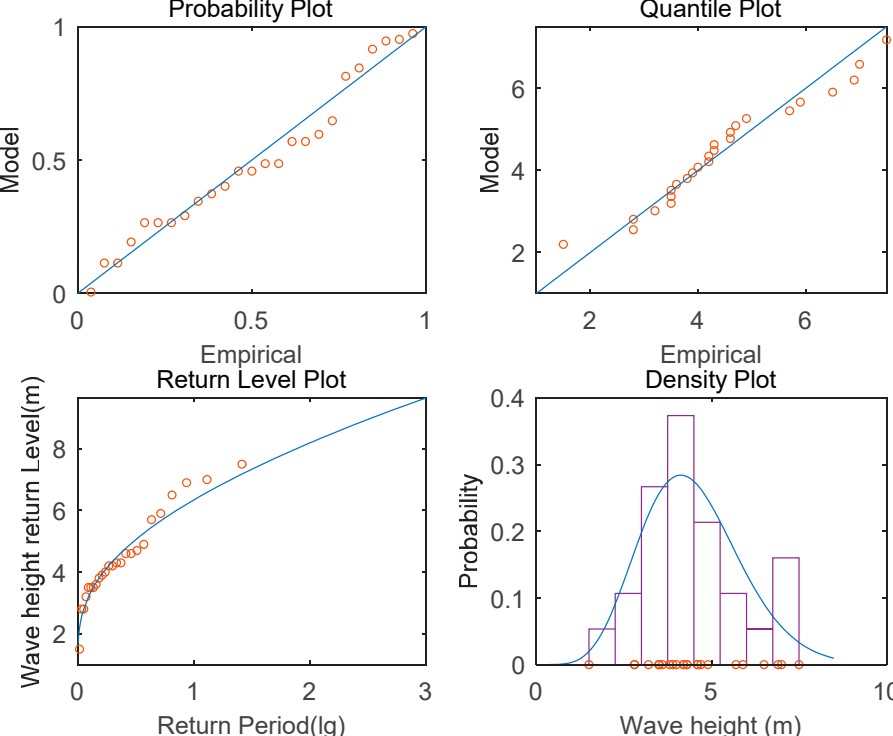

**Figure 6.** Inspection diagram of annual extreme wave height data of Maximum Entropy distribution.

**Table 4.** K-S test results and related parameters.

| Distribution Model | Related Parameters | Check Value $D_n$ | Critical Value $D_0(0.05)$ | Test Result |
|---|---|---|---|---|
| Gumbel | $\mu = 5.1952$ $\sigma = 1.4797$ | 0.2089 | 0.2641 | accept |
| Weibull | $a = 4.9600$ $b = 3.3680$ | 0.1542 | 0.2641 | accept |
| Pearson-III | $\alpha = 9.4155$ $\beta = 0.4733$ | 0.1138 | 0.2641 | accept |
| Maximum entropy | $\alpha = 0.0059$, $\beta = 0.1365$, $\gamma = 4.2220$, $\xi = 1.9368$ | 0.1074 | 0.2461 | accept |

**Table 5.** Statistical analysis results of annual extreme value method.

| Distribution Pattern | Design Wave Height/m | | | | | | |
|---|---|---|---|---|---|---|---|
| | 10a | 20a | 50a | 100a | 200a | 500a | 1000a |
| Gumbel | 6.34 | 7.15 | 8.21 | 8.99 | 9.78 | 10.81 | 11.60 |
| Weibull | 6.35 | 6.87 | 7.44 | 7.81 | 8.14 | 8.53 | 8.80 |
| Pearson-III | 6.39 | 7.08 | 7.92 | 8.51 | 9.07 | 9.79 | 10.31 |
| Maximum Entropy | 6.35 | 6.95 | 7.64 | 8.07 | 8.51 | 8.88 | 9.28 |

We use the four distribution patterns in Table 5 as the continuous distribution of the typhoon wave height in the traditional one-dimensional composite model. The discrete distribution is the Poisson distribution composed of the number of typhoons per year for analysis and calculation. The calculation results are shown in Table 6.

**Table 6.** Design wave height of traditional one-dimensional compound extreme value method.

| Distribution Pattern | Design Wave Height/m | | | | | | |
|---|---|---|---|---|---|---|---|
| | 10a | 20a | 50a | 100a | 200a | 500a | 1000a |
| Compound Gumbel | 7.65 | 8.47 | 9.53 | 10.32 | 11.10 | 12.14 | 12.92 |
| Compound Weibull | 7.15 | 7.57 | 8.04 | 8.35 | 8.64 | 8.98 | 9.22 |
| Compound Pearson-III | 7.39 | 7.97 | 8.67 | 9.17 | 9.64 | 10.24 | 10.68 |
| Compound Maximum Entropy | 7.28 | 7.80 | 8.39 | 8.73 | 9.13 | 9.39 | 9.50 |

It can be seen from Tables 5 and 6 that considering the annual frequency of typhoons and the distribution function of extreme wave heights, the compound extreme value distribution and the return value obtained are larger than the calculation result of the univariate extreme wave height distribution. Therefore, we show safety for the structure design. The calculated results of the design wave height of the composite maximum entropy distribution are between the composite Gumbel distribution and the composite Weibull distribution, and there is little difference between the calculated results of the composite Pearson III distribution. This shows that the calculation results of the composite maximum entropy distribution are reasonable and within the acceptable range. The maximum entropy-based distribution function is theoretically advanced and can reduce the a priori principle of the artificial assumed distribution function to some extent. At the same time, the distribution function contains four parameters, which can better fit the existing data and the calculated results are stable. Therefore, the composite maximum entropy distribution model has some advantages in the calculation of typhoon influence design wave height recurrence level.

The following uses different time scales to conduct a more in-depth analysis of the typhoon wave height. We take the number of typhoon occurrences within a certain period of time (one year) as the observation value, which is divided into $M_1$, $M_2$, $M_3$; $Y_1$, $Y_2$, $Y_3$ six samples. $M_1$ is a sample of the number of typhoons in summer, $M_2$ in autumn, and $M_3$ in winter; $Y_1$ is a sample of typhoons in July, $Y_2$ in August, and $Y_3$ in September. The number of typhoon occurrences at different time scales conform to the Poisson distribution.

The intensity of the Poisson distribution corresponding to different time scales for each sequence is shown in Table 7.

**Table 7.** Poisson intensity.

| Sequence | $M_1$ | $M_2$ | $M_3$ | $Y_1$ | $Y_2$ | $Y_3$ |
|---|---|---|---|---|---|---|
| Poisson strength $\lambda$ | 0.44 | 2.48 | 0.32 | 0.76 | 0.92 | 0.68 |

When the time scale $T$ is 3 months, we obtain samples $M_1$, $M_2$, and $M_3$ of typhoon occurrence times in summer, autumn, and winter, respectively, for the analysis and calculation, and the calculation results of design wave heights for different return periods are obtained through the new model (Table 8).

**Table 8.** Design wave height.

| Data Group | Design Wave Height/m | | | | | | |
|---|---|---|---|---|---|---|---|
| | 10a | 20a | 50a | 100a | 200a | 500a | 1000a |
| $M_1$ | 5.43 | 6.20 | 7.02 | 7.55 | 7.99 | 8.56 | 8.84 |
| $M_2$ | 7.08 | 7.62 | 8.24 | 8.61 | 8.87 | 9.33 | 9.46 |
| $M_3$ | 5.02 | 5.88 | 6.76 | 7.30 | 7.80 | 8.39 | 8.72 |

It can be seen from Table 8 that taking one year as the time period, the design wave height values obtained from the occurrence of typhoons in different seasons vary over time. Autumn typhoons occur most frequently, and the design wave height is higher, while in summer and winter, the corresponding design wave height is lower because of the fewer typhoons. Using the frequency of a once-in-a-hundred-year typhoon, for example, summer, autumn, and winter typhoon situations of the calculated design wave height values were 7.55 m, 8.61 m, and 7.30 m. The design wave height of autumn is higher than summer and winter, respectively, with 1.05 m and 1.31 m. For the frequency of a once in two hundred year typhoon, the autumn one is higher with 0.88 m and 1.07 m than in summer and winter, respectively. For the frequency of once-in-a-thousand-years, it is 0.63 m and 0.74 m higher in autumn than summer and winter.

When the time scale T is a one month period, we get the number of typhoons occurring in July, August, and September, the $Y_1$, $Y_2$, $Y_3$ to obtain the estimated result of the design wave height values for different return periods by the new model (Table 9).

**Table 9.** Design wave height.

| Data Group | Design Wave Height/m | | | | | | |
|---|---|---|---|---|---|---|---|
| | 10a | 20a | 50a | 100a | 200a | 500a | 1000a |
| $Y_1$ | 6.03 | 6.69 | 7.44 | 7.91 | 8.36 | 8.79 | 9.19 |
| $Y_2$ | 6.22 | 6.86 | 7.58 | 8.02 | 8.47 | 8.85 | 9.25 |
| $Y_3$ | 5.92 | 6.60 | 7.34 | 7.84 | 8.30 | 8.75 | 9.15 |

Taking one year as the time period, the design wave heights calculated from the occurrence of typhoons in different months vary over time. The most frequent occurrences of typhoons in autumn are mainly in July, August, and September. It can be seen from Table 9 that the design wave heights of the three samples are not much different. Taking the frequency of the once-in-a-hundred-year typhoon, for example, the calculated design wave height values of July, August, and September are 7.91 m, 8.02 m, and 7.84 m. August's value is higher than July and September with, respectively, 0.11 m and 0.18 m; July's value is higher than September's by 0.07 m. For a once in two hundred year typhoon, August is higher than July and September by 0.10 m and 0.17 m, respectively, and July is 0.06 m

higher than September. For a one in a thousand-year typhoon, August is only 0.07 m and 0.10 m higher than July and September, while July was only 0.04 m higher than September.

It can be seen from Figure 7 that taking the curve $M_2$ as an example, when the return period is 50 years, the design wave height value calculated by the new model is not much different from the design wave height value of the Gumbel distribution, which is 8.24 m and 8.21 m, respectively. The calculation result of the new model of design value for a once-in-a-hundred-year frequency is only 0.03 m higher than the calculation result of the Pearson-III distribution, which is 8.61 m and 8.58 m, respectively. Altogether, the calculation results of the new model are higher than those of the Weibull and Maximum entropy distributions and are also more stable. Since the impact of typhoons is taken into account, and autumn is when typhoons are most frequent, it is reasonable that the design wave height estimation result of the new model is higher than that of the traditional univariate extreme value model. Therefore, it can be valuable for coastal protection engineering safety to use the wave height design value obtained by the new model.

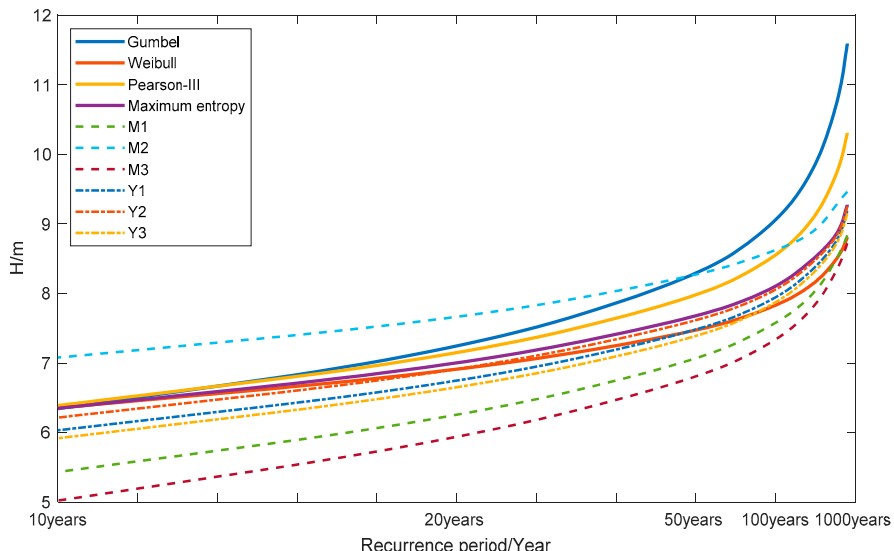

**Figure 7.** Comparison between the new model and the extreme value distribution.

From Figure 8, with the curve $M_2$ as an example, the design wave height value of the new model (for the frequency of a once in many year typhoon) is nearly in the middle of the design wave height of the compound Weibull distribution and the compound maximum entropy distribution. There is not a large difference between the two, which is lower than the compound Gumbel distribution and compound Pearson-III distribution design wave height values. The calculation result of the new model for the design value of the once-in-a-hundred-year wave height is 8.61 m, which is lower than the calculation results of the compound Gumbel distribution and the Weibull distribution, with a difference of 1.71 m and 0.56 m, respectively. The calculation results of the compound Weibull distribution and the compound maximum entropy distribution are 8.35 m and 8.73 m, respectively, which is similar to the calculation result of the new model. It can be seen that compared with the traditional one-dimensional compound extreme model, the annual typhoon frequency and annual extreme wave height are regarded as random variables to solve the return level, which makes the estimation results often too safe and somewhat wasteful. The new model takes into account the time factor and deeply analyzes the characteristics of typhoon intensity at different times, especially during frequent periods, and its influence on wave height, so as to show the statistical characteristics of design wave height in both time and space, making the calculation results more stable and accurate. On the basis of the safety of the structural design, it has a good economic value at the same time, achieving a good compromise between the two important aspects of "safety" and "economy" for marine engineering.

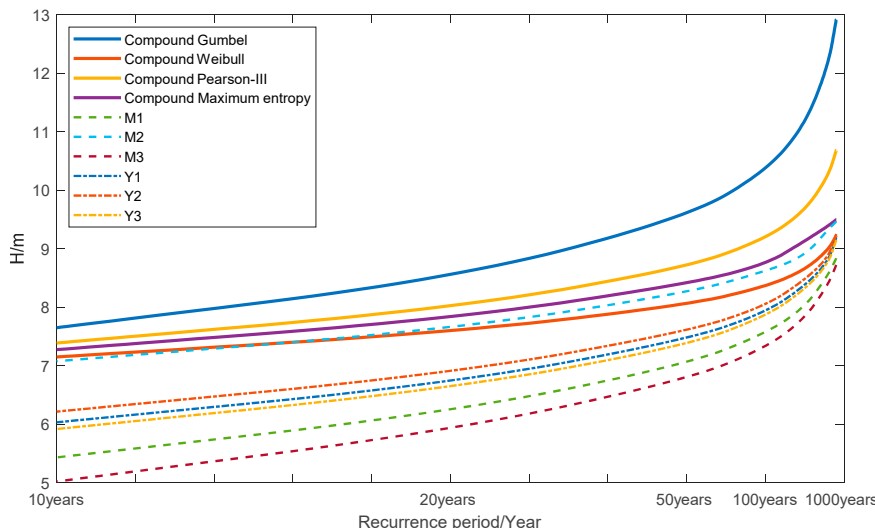

**Figure 8.** Comparison of the new model and the traditional compound distribution.

## 4. Conclusions

This paper introduces the stochastic process theory and the principle of maximum entropy in the study of typhoon wave heights and establishes a new model for typhoon design wave height estimation that includes the characteristics of time and space. The new model takes into account the distribution of typhoon wave height over time and can maintain the maximum uncertainty of the extension function of unknown information and reduce the interference of human factors so that the inherent statistical characteristics of typhoon wave height can be displayed more comprehensively; thus, providing a new method for the in-depth study of various marine environmental elements of typhoon waves. Based on the typhoon and wave height data of Naozhou, the return level of design wave height in different seasons and months is calculated. The new model is compared with the commonly used single variable extreme value model and one-dimensional composite extreme value distribution model, and the conclusions are as follows:

(1) Compared with the widely used univariate extreme value model, the distribution function with maximum entropy (derived based on the principle of maximum entropy) has its progressiveness in theory, which can reduce the a priori factors of artificially assumed distribution functions. Meanwhile, the distribution function contains four parameters, which can better fit the existing data, and the obtained calculation results are relatively stable. Thus, it has certain advantages in the calculation of the typhoon-affected return level for design wave height.

(2) Treating the frequency of typhoon occurrence as a Poisson process, with the consideration of the time course of typhoon occurrence, is more in line with the actual situation. The probability of each frequency of typhoon occurrence in Naozhou from June to September is predicted, and the results are basically in line with the actual situation. Therefore, based on the methods in this paper, we can reasonably predict the occurrence frequency of typhoons in the adjacent future from the existing typhoon data and make corresponding preventive measures to reduce or even eliminate the adverse effects of typhoon disasters on marine engineering structures.

(3) In the previous design wave height estimation methods, the wave height was regarded as a random variable, which could only reflect the statistical characteristics of the wave height in space. In this paper, the wave height is regarded as a random process, and the time perspective is added to the spatial angle to simultaneously reflect the wave height with space and time. With this improvement, it is feasible to conduct a more in-depth and comprehensive study on the law of wave height distribution. Ocean engineering (such as breakwaters or oil platforms) has huge economic costs, even if the design parameters are small. Using this method to analyze and calculate

the design wave height values in different seasons and months (during the typhoon occurrence period) can provide a more effective and accurate method to calculate seawall height in coastal protection projects to aid prevention and mitigation of typhoon surge disasters. It can also improve design parameter guidance to avoid economic losses caused by typhoon disasters.

In this paper, the design wave height estimation method based on the stochastic process and the principle of maximum entropy are theoretically advanced, which can provide new ideas and methods for the estimation of the return level for marine environmental elements under the influence of extreme weather. In future work, we can also calculate the joint return level of wind speed, wave height, and water increase under the influence of typhoons. In this study, we conduct a more comprehensive analysis and parameter design of multiple environmental factors under extreme sea conditions in marine engineering while providing more in-depth and comprehensive guidance for disaster prevention and mitigation.

**Author Contributions:** Methodology and Project administration, G.L.; Writing original draft preparation, P.X.; Software, Y.K.; Resources, F.W.; Investigation, Y.Y.; Data curation, D.Z.; Formal analysis and Writing—review and editing, Z.Y. All authors have read and agreed to the published version of the manuscript.

**Funding:** This research was funded by the National Natural Science Foundation of China (No. 52071306 and No.51379195), the Natural Science Foundation of Shandong Province (No. ZR2019MEE050 and No. U1806227), the 111 Project (No. B14028), and the Graduate Education Foundation (No. HDYA19006).

**Institutional Review Board Statement:** Ethical review and approval were waived for this study, due to this study not involving humans or animals.

**Informed Consent Statement:** Patient consent was waived due to this study not involving humans.

**Data Availability Statement:** The data presented in this study are available on request from the corresponding author. The data are not publicly available due to they are private data that we buy.

**Conflicts of Interest:** The authors declare no conflict of interest.

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
