# Peer review of "Design Wave Height Parameter Estimation Model Reflecting the Influence of Typhoon Time and Space"

_jmse, doi:10.3390/jmse9090950_

Round 1
Reviewer 1 Report
The authors provided a thorough revision and the revised manuscript clearly articulates the main takeaways of this work. I have a few minor comments.
Please go through the manuscript to correct spells and word spacing.
Line 76 & 81– Please mention the full name of AM, GPD, POT, ECMFW
Line 90-92 - Yes. As threshold value selection in the POT method is subjective and can introduce user bias, some studies determined a threshold value that provides the best fit to the modeled GPD using quantile-quantile plots to avoid the user bias (e.g., Neary et al. [15]). Please review the POT method used in [15].
Figure 2 – Is this monthly maximum wave height?. Confusing. Some years have Zero wave height. Why?
Author Response
Response to Reviewer 1 Comments
Thank you for the reviewer’ comments concerning our manuscript. We do appreciate your comments on our manuscript. Those comments are all valuable and very helpful for revising and improving our paper, as well as the important guiding significance to our researches. We have studied comments carefully and have made correction which we hope meet with approval. Revised portion are marked in red in the paper. Your support and help are wholeheartedly expected on the future developing road of our work. Now, the explanation for the issues raised in the review comments has been listed as follow.
Point 1:Please go through the manuscript to correct spells and word spacing.
Response 1: Thank you for your suggestions. We carefully checked the spelling and word spacing of the full text, and made the following modifications and marked them in the text:
Line 320-321, 334 -336: There is no space in the middle of "mand". It should be "m and".
Line 372: " int roduces" should be "introduces".
Point 2:Line 76 & 81– Please mention the full name of AM, GPD, POT, ECMFW
Response 2: Thank you for reading the article carefully. We add the full names of AM, GPD, POT and ECMFW, and the specific modifications are as follows:
Line78: annual extreme value method (AM)
Line79: generalized Pareto distribution (GPD)
Line79: peak over threshold (POT) method
Line86: European Center for Medium-Range Weather Forecasts (ECMWF)
Point 3:Line 90-92 - Yes. As threshold value selection in the POT method is subjective and can introduce user bias, some studies determined a threshold value that provides the best fit to the modeled GPD using quantile-quantile plots to avoid the user bias (e.g., Neary et al. [15]). Please review the POT method used in [15].
Response 3: Thank the reviewer for your attention and professional explanation on this issue. We carefully reviewed the pot method used in [15], the following description is supplemented:
Line 81: By using quantile-quantile plots to determine the threshold for GPD distribution fitting is indeed a great method to avoid users deviation.
And in the future related research work,We will pay attention to this method of determining the threshold and apply it to the calculation of design parameters of marine environmental elements.
Point 4:Figure 2 – Is this monthly maximum wave height? Confusing. Some years have Zero wave height. Why?
Response 4: Thank you for reading the article carefully. The data in Figure 2 is the sorted and screened data, which represents the monthly maximum wave height from June to September during the typhoon occurrence of each year. In some years, the value of zero indicates that there is no typhoon in a month of that year, so its data value is zero. We have added the following contents in the text:
Line 253: In some years, the value of zero indicates that there is no typhoon in a month of that year

Reviewer 2 Report
I have read with interest the paper by Liu et al. and I have not raised criticisms that may impend its publication. I have only a few minor comments, which I encourage to consider. - Abstract. In the first paragraph, it is introduced the storm surge concept, which is not considered in the paper that instead focuses on the wave heights. - Abstract: how can authors prove the model they propose is more accurate? - Lines from 154. Define all the variables when they are used for the first time. - Eq 4. Use not bold characters. - Line 201: How many months are authors considering? State the correct number that is related to the Poisson’s parameter. - Line 203. The link points to a Chinese website. Please share a link that might be also used by no-Chinese people. - Figure 2. Explain the variables shown in the plot. Wave height should be Significant wave height. - All Tables displaying wave height data. Going up to 1/10 mm is meaningless for practical applications. I suggest stopping the digits to cm, also throughout the text. - Here and there, spaces are missing between words or are added within a word. Please check.Author Response
Response to Reviewer 2 Comments
Thank you for the reviewer’ comments concerning our manuscript. We do appreciate your comments on our manuscript. Those comments are all valuable and very helpful for revising and improving our paper, as well as the important guiding significance to our researches. We have studied comments carefully and have made correction which we hope meet with approval. Revised portion are marked in red in the paper. Your support and help are wholeheartedly expected on the future developing road of our work. Now, the explanation for the issues raised in the review comments has been listed as follow.
Point 1:Abstract. In the first paragraph, it is introduced the storm surge concept, which is not considered in the paper that instead focuses on the wave heights.
Response 1: Thank you for reading the article carefully. Extreme sea conditions are induced by typhoon storm surge. The data used for statistical analysis of extreme wave height in this paper are based on the measured data under storm surge environment. The calculation of design wave height recurrence level is an important parameter of coastal and marine engineering design standards. It is one of the important tasks of disaster prevention and reduction in coastal areas of the world to reasonably and accurately calculate the design wave height of typhoon affected sea areas, predict and resist typhoon storm surge disasters. This paper focuses on the statistical characteristics of typhoon wave height, so as to provide more effective and accurate parameter guidance for coastal engineering design, so as to give consideration to the safety protection of typhoon storm surge disaster and the economy of engineering construction.The specific modifications are as follows:
Line 15: It is one of the important tasks of disaster prevention and reduction in coastal areas of the world to reasonably and accurately calculate the design wave height of typhoon affected sea areas, predict and resist typhoon storm surge disasters.
Point 2:Abstract: how can authors prove the model they propose is more accurate?
Response 2: Thank you for reading the article carefully. We have made the following explanations for the reviewer's question:
The traditional calculation of design wave height in the sea area affected by typhoon takes typhoon and wave height as random variables to solve the recurrence level. Its sample is a set of one-dimensional time series and a sample function of random process. In this paper, typhoon and wave height are regarded as random processes, and the statistical characteristics of typhoon wave height are studied from the perspectives of time and space. At the same time, the dimension of the analysis covers the previous analysis of marine environmental factors based on random variables. The results are more in line with the natural real state of typhoon wave height both in theory and in engineering practice. At the same time, the design wave height distribution model determined based on the maximum entropy method can better fit the existing data. Therefore, compared with the traditional methods and models. This article has its advanced theory and the conclusion is more comprehensive and reasonable. The corresponding amendments and supplements are as follows:
Line 17: In this paper, the design wave height estimation method based on the stochastic process and the principle of maximum entropy is theoretically advanced, and it can provide a new idea as well as a new method for the estimation of the return level for marine environmental elements under the influence of extreme weather.
Point 3:Lines from 154. Define all the variables when they are used for the first time.
Response 3: Thank you for reading the article carefully. We defined all the variables when they are used for the first time. The specific modifications are as follows:
Line 168: Wherein is the number of typhoons, ,,, are the parameters given by the constraints and boundary conditions.
Point 4:Eq 4. Use not bold characters.
Response 4: Thank you for reading the article carefully. We have removed bold for Formula 4.
Point 5:Line 201: How many months are authors considering? State the correct number that is related to the Poisson’s parameter.
Response 5: Thank you for reading the article carefully. We considered a total of four months of typhoons from June to September. Typhoons in our study area generally occur from May to September. In fact, based on our typhoon data, the number of typhoons in May is very small. Therefore, we consider the probability of different times of typhoons from June to September when typhoons do not occur in May. At this time, the Poisson parameter is actually the average number of typhoons from June to September. The relevant modification is as follows:
Line 211: the frequency of typhoons and the corresponding Poisson process intensity in Naozhou from June to September in 1990-2016 are listed in Table 1.
Point 6:Line 203. The link points to a Chinese website. Please share a link that might be also used by no-Chinese people.
Response 6: Thank you for your advice. This website is the source of our typhoon data, there is no no-Chinese website that can obtain the same data. But we will pay more attention to this problem in the future research work to facilitate readers' reading.
Point 7:Figure 2. Explain the variables shown in the plot. Wave height should be Significant wave height.
Response 7: Thank you for reading the article carefully. We have supplemented the description of wave height data type, and the specific modifications are as follows:
Line 251: Figure 2 shows the 25-year significant wave height data set from June to September of Naozhou from 1990 to 2016 (missing 2004 and 2007).
Point 8:All Tables displaying wave height data. Going up to 1/10 mm is meaningless for practical applications. I suggest stopping the digits to cm, also throughout the text.
Response 8: Thank you for your suggestion and we accept it completely. We have stopped the digits to cm throughout the text, and mark them in the text.
Point 9:Here and there, spaces are missing between words or are added within a word. Please check.
Response 9: Thank you for reading the article carefully. We carefully checked the full text and made corresponding modifications, which are marked in the text.
